# Simulation of Air Puff Tonometry Test Using Arbitrary Lagrangian–Eulerian (ALE) Deforming Mesh for Corneal Material Characterisation

**DOI:** 10.3390/ijerph17010054

**Published:** 2019-12-19

**Authors:** Osama Maklad, Ashkan Eliasy, Kai-Jung Chen, Vassilios Theofilis, Ahmed Elsheikh

**Affiliations:** 1School of Engineering, University of Liverpool, Liverpool L69 3GH, UK; 2NIHR Biomedical Research Centre for Ophthalmology, Moorfields Eye Hospital NHS Foundation Trust and UCL Institute of Ophthalmology, London EC1V 9EL, UK; 3School of Biological Science and Biomedical Engineering, Beihang University, Beijing 100191, China

**Keywords:** Ocular biomechanics, Intraocular pressure (IOP), Fluid–Structure Interaction (FSI), Arbitrary Lagrangian–Eulerian mesh (ALE)

## Abstract

Purpose: To improve numerical simulation of the non-contact tonometry test by using arbitrary Lagrangian-Eulerian deforming mesh in the coupling between computational fluid dynamics model of an air jet and finite element model of the human eye. Methods: Computational fluid dynamics model simulated impingement of the air puff and employed Spallart–Allmaras model to capture turbulence of the air jet. The time span of the jet was 30 ms and maximum Reynolds number was$\;R_{e}=2.3\times 10^{4}$, with jet orifice diameter 2.4 mm and impinging distance 11 mm. The model of the human eye was analysed using finite element method with regional hyperelastic material variation and corneal patient-specific topography starting from stress-free configuration. The cornea was free to deform as a response to the air puff using an adaptive deforming mesh at every time step of the solution. Aqueous and vitreous humours were simulated as a fluid cavity filled with incompressible fluid with a density of 1000 kg/m^3^. Results: Using the adaptive deforming mesh in numerical simulation of the air puff test improved the traditional understanding of how pressure distribution on cornea changes with time of the test. There was a mean decrease in maximum pressure (at corneal apex) of 6.29 ± 2.2% and a development of negative pressure on a peripheral corneal region 2–4 mm away from cornea centre. Conclusions: The study presented an improvement of numerical simulation of the air puff test, which will lead to more accurate intraocular pressure (IOP) and corneal material behaviour estimation. The parametric study showed that pressure of the air puff is different from one model to another, value-wise and distribution-wise, based on cornea biomechanical parameters.

## 1. Introduction

Biomechanical properties of biological tissues are important health indicators and multiple clinical decisions and surgical planning can be made based on their dynamic response to loading [1]. However, some of the mechanical and dynamic responses are still not fully understood due to the non-linearity and viscoelasticity of the tissues [2]. The tissue of interest in this study is the cornea, which contributes significantly to the optical focusing power of the eye and is a vital area in refractive surgeries [3,4,5]. The air puff test conducted by the Ocular Response Analyser (ORA; Reichert, Inc., Buffalo, NY, USA), the CorVis-ST (Oculus Optikgerate GmbH, Wetzlar, Germany) [6,7,8], etc. is a non-contact tonometry method with direct interaction with the cornea used to estimate intraocular pressure that is necessary for glaucoma management [9,10,11]. Moreover, this test was proven to have a promising potential in corneal material characterisation and Keratoconus detection [12,13,14,15]. Tonometry is based on a simple concept of applying a certain load causing deformation in the cornea and relating the deformation parameters to the pressure inside the eye. However, accuracy of intraocular pressure (IOP) estimation continues to be a challenge due to the effect of corneal parameters including corneal geometry and material properties [16,17]. The interplay between corneal geometry, material properties, ocular fluids and the air puff was studied before theoretically, numerically and clinically but with assumptions for the fluid–structure interaction (FSI) effect. 

Theoretically, the air puff test was simulated as a harmonic oscillator model (1-DOF) to model behaviour of the cornea under action of the air puff test by Zhaolong, et al. [18]. They investigated the air puff induced corneal vibrations and their effect on intraocular pressure (IOP), viscoelasticity and mass of the cornea based on theoretical approach and some clinical observations. Moreover, Pandolfi et al. [19] used two different approaches to estimate intraocular pressure and the other eye parameters; the first approach was modelling the corneal system as a harmonic oscillator and, in the second approach, they used patient specific geometries and finite element models to simulate the dynamic test on surgically treated corneas. The finite element calculations reproduced the observed clinical deformations of cornea including the two applanation configurations provided by Ocular Response Analyzer, suggesting that the mechanical response of cornea to the air puff test was driven only by elasticity of the stromal tissue. Furthermore, Kaneko et al. [20] modelled the human eyeball as a 1-DOF and 2-DOF systems to assess the dynamic response of the cornea and eyeball to the air puff test, as shown in Figure 1. 

Numerically, Kling et al. [21] presented a two-dimensional axis-symmetric finite element model which predicts deformation patterns of the cornea during air puff test to get its elastic and viscoelastic properties. They validated the results against experimental testing on porcine and human eyes to get the spatial pressure profile. They developed a 2D axis-symmetric CFD model for the air jet impinging on different solid configurations of the cornea. Their parametric study revealed significant contributions of intraocular pressure and corneal thickness to the corneal deformation, besides the corneal biomechanical properties [21]. Moreover, a patient specific finite element model of a healthy eye was presented by Gracia et al. [22], taking account of the stress free configuration. The cornea was modelled as an anisotropic hyperelastic material with two preferential directions. Three sets of parameters within the healthy human range, based on inflation tests, were considered. A two-dimensional CFD simulation of the air jet was used to obtain pressure loading exerted on the anterior surface of the cornea; however, the cornea was considered a solid non-deformable surface.

In another study, Muench et al. [23] identified the normal and shear stress profiles on cornea resulting from an air puff to present a universal equation of the pressure distribution on cornea to use it for corneal material inverse analyses. Their method was based on experimental characterisation of the air puff produced by CorVis-ST and CFD simulation of the air puff test. As a calibration of the CFD simulations, they applied the air puff to a rigid eye model which was hung up through a yarn and positioned in front of the nozzle exit. They used eleven corneal deformation configurations to apply them in the CFD model, but also the cornea was simulated as a rigid surface. The outcomes showed dependency of pressure distribution on cornea on corneal deformations with minor effect of shear stress component on corneal deformations. To add a realistic modelling of the human eye, they considered the human face to see its influence on the pressure distribution on cornea. They demonstrated that pressure and shear stress distributions were not rotationally symmetric when applying the air puff to real human eyes [23]. 

Furthermore, using mesh-free particle method, Montanino et al. [24,25] proposed the first 3D fluid–solid interaction model between cornea and aqueous humour under the air puff test. Their numerical results confirm the importance of including the internal fluids in simulation of the non-contact tonometry. However, they considered a cornea only model and applied an analytical bell-shaped pressure distribution over the cornea with assumptions on the interaction between the air-puff and cornea. The closest fluid–structure interaction simulation of the non-contact tonometry test was presented by Garcia et al. [26], motivated by the fact that the proper interaction between the air and cornea is still unknown. They explored four different approaches starting from structural analysis to considering the fluid–structure interaction with the air puff from outside and with the aqueous humour from inside. However, the model was created based on 2D-axisymmetric porcine eyes. The results indicate the importance of considering fluid–structure interaction effect on the pressure distribution and corneal deformations, which will lead, if not considered, to an overestimated IOP measurements and biased corneal stiffness when performing the inverse finite element analysis [27]. To the best of our knowledge, the current study is the first attempt to quantify the influence of fluid–structure interaction from the air puff on corneal behaviour predictions for 3D corneal patient-specific eye models using the Arbitrary Lagrangian–Eulerian (ALE) deforming mesh, with more focus on air puff dynamics and extending the model for a larger parametric study aiming to develop IOP and corneal material estimation algorithms.

The impinging air puff is commonly studied assuming round jet diffusion and using impingement theory [28,29]. In this theory, flow characteristics of impinging jets depend on parameters including jet orifice diameter, nozzle to impingement surface distance, jet confinement, radial distance from stagnation point, angle of impingement, surface curvature and roughness, nozzle exit geometry and turbulence intensity [30,31,32]. The round jet is characterised by a continuous increase in thickness of boundary shear layer. This boundary layer has two corresponding factors: a decrease in jet core cross-section and an increase in jet diameter as shown in Figure 2a. The core length depends on the inner angle of diffusion, about 5° for the jet core and around 8.5° for the outer jet diameter for highly turbulent impinging jets [33]. Figure 2b shows three regions of an impinging jet: the “free” jet region, the impingement or stagnation region and the wall-jet region. In an earlier study, Larras considered the free and impinging jet regions and provided a detailed analysis of plane turbulent impinging jets [33]. The paper is arranged as follows. Section 2 states materials and numerical methods used in the analysis. Section 3 presents some of the achieved results. Section 4 provides a discussion about methods and results.

## 2. Materials and Methods

The numerical model of the air puff test, shown in Figure 3, was constructed as a coupled model between computational fluid dynamics (CFD) and computational solid dynamics (CSD) as implemented in the software package ABAQUS (version 6.14, Dassault Systemes Simulia Inc., Providence, RI, USA). The air puff test simulation consisted of three components: Three-dimensional finite element model of the eye and material models for ocular tissuesThree-dimensional CFD turbulence model of the air puff impinging on the corneaFluid–structure interaction (FSI) coupling between the two models

### 2.1. Three-Dimensional Eye Model

The eye model consisted of 10,000 fifteen-nodded continuum elements (C3D15H), a general-purpose element with nine integration points, arranged over two layers, distributed along 15 rings in the cornea and 35 rings in the sclera. This number of mesh rings was the outcome of a mesh dependence study which is available in the Appendix A. The corneal topography was based on the Pentacam measurements of topography and thickness profile. For sclera, we do not have information on the accurate dimensions specific to each patient. Hence, we used idealised geometry for the sclera based on average clinical dimensions; however, depending on age of the patient, the scleral material stiffness was adjusted. Therefore, by patient specific eye models, we mean the material of the whole eye globe is specific to the patient and adjusted based on age. However, the geometry of sclera is not specific to the patient and relies on idealised average values [34,35]. The model also incorporated attributes to represent in vivo conditions including the non-uniform thickness of cornea and sclera, weak inter-lamellar adhesion in corneal stroma, and asphericity of the cornea’s anterior and posterior surfaces [36].

The eye model included five different material definitions for cornea, limbus, anterior, equatorial and posterior sclera behaving hyperelastically and their stiffness increases gradually under loading following an S-shaped stress–strain path, as reported previously in experimental studies [35,37,38]. With these important features, the model can select which stress–strain path (under loading or unloading) that each element would follow based on its strain history. The hyperelastic materials have a rubber-like material behaviour and the strain energy potential function (∏) is different from elastic materials and it takes multiple forms. The Ogden form in Equation (1) was the one applied in the finite element model of the human eye [36,39,40].
(1)$$\prod =\;\sum_{i=1}^{N}\frac{2\mu_{i}}{\alpha_{i}{}^{2}}\left({\bar{\lambda_{1}}}^{\alpha_{i}}+{\bar{\lambda_{2}}}^{\alpha_{i}}+{\bar{\lambda_{3}}}^{\alpha_{i}}-3\right)+\sum_{i=1}^{N}\frac{1}{D_{i}}{\left(J^{\mathrm{el}}-1\right)}^{2i}$$
where $\bar{\lambda_{i}}$ are the deviatoric principal stretches, which are related to the deformations at constant volume as outcome of shear stresses; N defines order of the Ogden model with maximum of sixth order (N = 6); μ and α are material parameters; $J^{\mathrm{el}}$ is the elastic volume ratio related to thermal expansion; and $D_{i}$ defines the material compressibility.

Finite element model of the eye was prevented from rigid body motion in the Z-direction (anterior–posterior) at the equatorial nodes. In addition, the posterior and anterior pole nodes were restricted in X and Y directions, to prevent rotation, but were free to move in the Z-direction (anterior–posterior) (see Figure 3). The rest of the nodes in the model were free to move in all directions. Before analysis, the stress-free geometry of the eye was estimated. It is important to calculate the un-deformed configuration of the eye before applying the IOP, since deformed geometry of the eye will not be suitable for applying different values of IOP when performing the parametric study. An iterative approach was used to gradually move the mesh nodes to reach the stress-free (relaxed) configuration of the ocular tissue [36,41]. An initial numerical model was generated based on a measured value of IOP and corneal patient-specific geometry of the ocular tissue. The calculated deformations were then subtracted using inverse calculations from the stressed geometry to get the relaxed (stress-free) geometry.

To represent intraocular pressure inside the eye, the fluid cavity technique was used. This technique was mainly used to simulate fluid-filled structures such as pressure vessels, hydraulic or pneumatic actuators and automotive air bags. The fluid cavity behaviour governs the relationship between cavity pressure, structure deformation and volume [35,42,43]. The fluid cavity calculates the change in IOP and internal volume during application of the air puff and corneal deformations. The fluid cavity was filled with a fluid with a density of 1000 kg/m^3^ and bulk modulus of $2.2\times 10^{3}$ MPa [44]. A reference node was specified inside the cavity to represent the applied pressure and used in the volume calculations. Even though there are multiple components inside the eye including ocular lens, iris, aqueous and vitreous, the human eye was dealt with as a deformable pressure vessel that has internal pressure equal to IOP applied using a fluid cavity taking into account the change in pressure with the change in volume. All the eye models were generated using a built-in-house software package [38] with control on geometrical and material parameters.

### 2.2. Three-Dimensional CFD Turbulence Model of the Air Puff

The CFD code in Abaqus/CFD was validated in our previous work against Laser Doppler Anemometer (LDA) data for an impinging jet on a flat surface [45]. Model of the air puff consisted of 103,680 six-nodded 3-D fluid continuum elements (FC3D6) and used Spalart–Allmaras turbulent eddy viscosity model [46,47] to simulate the turbulence in the air jet. The air model domain and mesh were created over the cornea and a 4 mm ring of the sclera by projecting coordinates of the anterior surface nodes to a distance of 11 mm from the cornea apex, as shown in Figure 3. The projection principle was based on the concept of similar triangles to calculate new coordinates of the air domain as projected from the eye coordinates. It was important to generate a MATLAB® code which was applicable for all eye geometries, idealised and corneal patient-specific, healthy or with certain pathological conditions. Material properties of air were defined in terms of density 1.204 kg/m^3^ and dynamic viscosity $1.83\times 10^{-5}$ kg·s^−1^/m [48,49]. Amplitude of the air jet velocity, and its variation with time were defined according to Figure 4a based on experimental data obtained from the manufacturers of CorVis-ST (Corneal Visualization Scheimpflug Technology) and based on a simulated CFD model of the air flow inside the device starting from the piston all the way to the nozzle [50] (see the Appendix A). The initial turbulent kinematic eddy viscosity was defined as four times the air kinematic viscosity $68\times 10^{-6}$ m^2^/s [42,49]. The CFD solution parameters were then specified in terms of momentum, pressure and transport equation solvers and which turbulence model to be used to resolve the turbulent fluctuations. In the CFD model, the air jet inlet diameter was set to 2.4 mm, as measured for the nozzle of CorVis-ST, and the air maximum velocity at the inlet was set to 167.8 m/s. The surface that surrounds the jet diameter was set as a no-slip wall boundary condition and the side was open to the air with gauge pressure equal to zero. Lastly, the cornea and 4 mm ring from sclera were set to a co-simulation, data exchange interface.

### 2.3. Fluid–Structure Interaction Co-Simulation

In the fluid–structure coupled analysis (videos are in the Appendix A), the eye finite element model imported the forces and lumped mass from the CFD model and exported deformations and velocities back to the CFD model at every time step (tn) of the job, as shown in Figure 4b. The structure model calculates deformations (Line 1) and sends them to the fluid model (Line 2) which in turn calculates pressure loads (Line 3) and returns them back to the structure model (Line 4). For the co-simulation step to run successfully, the interaction surfaces in the eye and air models should be exactly the same with the same node numbering. The air puff test is a transient problem, and Abaqus/CFD used an advanced second-order projection method to create an arbitrary deforming mesh [51]. It used node-centred finite-element discretisation for the pressure and a cell-centred finite volume discretisation of all other transported variables (such as velocity, temperature, turbulence, etc.) [42]. This hybrid meshing approach removed the need for any artificial dissipation, while preserving the traditional conservation properties associated with the finite volume method. The parallel preconditioned Krylov solvers (DSGMRES-ILUFGMRES) [52,53,54] were the main solution methods for transport equations including momentum and turbulence with prescribed iteration limit and convergence criteria (see Appendix B). The pressure and distance function equations were solved with one of the Krylov solvers (diagonally-scaled FGMRES, built-in Abaqus®), and strong multigrid algebraic preconditioner [51,55,56], parameters are given in Appendix B. 

The time was integrated using second-order accuracy and all other diffusive and advective terms were integrated using the Crank–Nicolson method [57]. The CFL (Courant–Friedrichs–Lewy) stability condition was also satisfied by continually adjusting the time increment size. The maximum value for CFL number was kept at 0.45. The CFL condition was necessary for partial differential equations’ solution convergence [58,59]. It must be less than one for explicit solvers to converge since the full numerical domain of dependence must contain the physical domain of dependence such as Laney’s definition [60].

#### Arbitrary Lagrangian–Eulerian (ALE) Deforming Mesh

To prevent distortion of the fluid mesh in the air puff model, the adaptive Arbitrary Lagrangian–Eulerian (ALE) deforming mesh was used. In FSI applications, where there are large solid deformations, the adaptive mesh is important for stability of the solution [61,62,63]. This method has the following characteristics: the mesh motion is constrained only at the free boundaries but everywhere else the material and mesh motion are independent. Moreover, it incorporates two main tasks: creating a new mesh and remapping the solution variables, through a process named advection, from the old mesh to the new mesh [61,62,63,64]. The meshing was created at a pre-specified frequency accompanied by a combination of mesh smoothing methods [42]. Then, the solution variables were remapped to the new mesh with second-order accuracy and conserving mass and momentum. For FSI stabilisation, the solution control parameters used to maintain the mesh quality and motion control are given in Appendix B. The adaptive deformable mesh for a quarter model of the air puff model is shown in Figure 5a, showing the initial corneal highest concavity meshes. Figure 5b illustrates the benefit of using ALE deforming mesh in capturing the FSI effect between the air puff and the cornea and comparing the apical deformation against finite element only model where the air pressure was put as a fixed value with a certain amplitude with time, as suggested by previous studies to be 50% of the piston pressure reading [6,65]. 

### 2.4. Clinical Dataset

A clinical dataset of 476 healthy participants from the Vincieye Clinic in Milan, Italy and Rio de Janeiro Corneal Tomography and Biomechanics Study Group, Brazil, was used to validate the numerical model. Institutional review board (IRB) ruled that approval was not obligatory for this record review study. However, the ethical standards as set in the 1964 Declaration of Helsinki, and revised in 2000, were observed. All participants provided informed consent before using their data in the study. All participants had a complete ophthalmic examination, including the CorVis ST (SW version 1.2r1307) and Pentacam (OCULUS Optikgeräte GmbH; Wetzlar, Germany). The inclusion criteria of healthy subjects were a Belin/Ambrósio Enhanced Ectasia total deviation index (BAD-D) of less than 1.6 the standard deviation (SD) from normative values in both eyes, no previous ocular surgery and systematic conditions including diabetes, myopia less than 10D and no concurrent or previous glaucoma or hypotonic therapies [66]. Moreover, to confirm the diagnosis, all exams of each clinic were blindly re-evaluated by a corneal expert at the other clinic. 

Cornea biomechanical response parameters were collected from the CorVis-ST including maximum deformation, applanation pressures and times, highest concavity, spatial and temporal deformations with age ranging 10–87 years, central corneal thickness 455–630 μm and IOP 9–25 mmHg. Table 1 provides descriptive statistics of the clinical dataset for the centres in Milan and Rio.

Corneal patient-specific numerical eye models were produced using an in-house MATLAB code to perform a parametric study with wide range of CCT, IOP and corneal material properties starting from the stress-free geometry. Their deformation patterns, as a response to the air puff, were analysed and compared against the clinical behaviour. 

### 2.5. Ethical Statement

A clinical dataset of 476 healthy participants from the Vincieye Clinic in Milan, Italy and Rio de Janeiro Corneal Tomography and Biomechanics Study Group, Brazil, was used to validate the numerical model. Institutional review board (IRB) ruled that approval was not obligatory for this record review study. However, the ethical standards as set in the 1964 Declaration of Helsinki, and revised in 2000, were observed. All participants provided informed consent before using their data in the study. All participants had a complete ophthalmic examination, including the CorVis-ST and Pentacam (OCULUS Optikgeräte GmbH; Wetzlar, Germany) exams. The inclusion criteria of healthy subjects were a Belin/Ambrósio Enhanced Ectasia total deviation index (BAD-D) of less than 1.6 the standard deviation (SD) from normative values in both eyes, no previous ocular surgery and systematic conditions including diabetes, myopia less than 10D and no concurrent or previous glaucoma or hypotonic therapies [66]. Moreover, to confirm the diagnosis, all exams of each clinic were blindly re-evaluated by a corneal expert at the other clinic.

## 3. Results

### 3.1. Air Puff Traverses

The air puff was analysed to see change of the velocity, pressure and mesh deformation during the test. Figure 6 shows two velocity components of the air puff, the axial velocity (V3) normal to the cornea and velocity component (V1) parallel to the cornea at three normal traverses (Y/D = 0, Y/D = 1, Y/D = 2), shown in Figure 3, and four time steps T= 5, 8, 10, 16 ms. By the time the puff gets stronger to reach its maximum strength at T = 16 ms and as the distance from the puff orifice increases, the normal velocity decreases until it reaches zero at the stagnation point on the cornea surface. By changing the path or the axial traverse further away from the cornea centre, the puff gets weaker and is noticed at (Y/D = 1 and 2), there are some negative values for the normal velocity indicating reflection of the air from cornea surface in the opposite direction to the flow. The jet accelerates parallel to the cornea forming a radial wall jet, developing with time and going further from the cornea centre axis. This explains why there is a negative pressure observed at this location of the cornea. The pressure was found to change with corneal deformations and time steps, as illustrated in Figure 7a. It is noticed from the plots that the distance from the jet at the end of every curve is increasing because of the movement of cornea with time of the test. Pressure here represents the static pressure; it starts with zero at the jet orifice and increases gradually towards the cornea because of transforming the dynamic pressure into static pressure. FSI was found to have an effect on the pressure distribution on cornea during time of the air puff test. Figure 7b shows the pressure distribution change with time and the region where there is negative pressure. Figure 7c shows the progression of corneal deformation with time, while Figure 7d indicates the difference between taking the FSI effect into account and ignoring it, through considering the cornea as a rigid, non-deformable, surface. Two different simulations of the turbulent jet were performed; one impinging on a rigid corneal surface with no moving boundaries and the other using FSI coupling between air and eye models to consider corneal deformations corneal surface with no moving boundaries and the other using FSI coupling between air and eye models to consider corneal deformations.

### 3.2. Parametric Study Results

A parametric study was done on the coupled model of the air puff test by changing four parameters of the eye model and simulating response of the cornea to the air puff, which gives a great understanding of how corneal biomechanical parameters affect its deformation, which in turn, affect IOP measurement and corneal material estimation. The four parameters involved in the study were:Cornea material stiffness coefficient (μ)Central corneal thickness (CCT)Corneal curvature radius (R)Intraocular pressure (IOP)

The total number of models included in the study was 110 models with wide ranges for CCT, IOP, R and corneal material coefficient (μ) representing the change in corneal stiffness using the built-in-house software package to generate the human eye model [38] and the newly generated MATLAB® code to build the air domain. Figure 8 shows the influence matrix of IOP and corneal material on dynamic corneal response parameters with different colour for each value. In Figure 8a, six corneal response parameters are plotted against each other and at different levels of IOP. The first row for the highest concavity (HC) deformation is the most explaining parameter. By increasing IOP, the HC deformation is lower. The opposite is happening with the stiffness parameter (SP-HC), Equation (2): for higher IOP, the stiffness parameter is higher. In Figure 8b, by increasing the stiffness of corneal material, the amount of deformation decreases and the peak distance (PD) between the applanation points shows the same trend. Figure 8c illustrates the corneal profile stages from initial geometry to highest concavity.
(2)$$\mathrm{SP}-\mathrm{HC}\;=\frac{\mathrm{AP}1-\mathrm{IOP}}{\mathrm{HC}\;\mathrm{deformation}-A1\;\mathrm{deformation}}$$

After showing, graphically, the influence of the parameters involved in the parametric study, it was vital to quantify correlations and significance of relationships between parametric study’s input and output parameters, to choose which response parameters were influenced more by changing IOP and corneal stiffness. This was an important outcome of the present study, as estimation algorithms for IOP and corneal material behaviour are required to correct FSI effect between the air puff and human cornea. A bivariate correlation analysis using SPSS statistics (version 24, IBM Corp., Armonk, NY, USA) was performed to obtain Pearson’s correlation coefficient (r) and two-tailed significance t-test to know the significance level of correlations (P-value). Descriptive statistics of the parametric study are shown in Table 2, providing mean, standard deviation, minimum and maximum of input and output parameters for 110 different eye models. 

Table 3 provides values of Pearson’s correlation coefficient (r) between input and output response parameters, which gives an indication on correlation’s strength and direction. The highest correlated parameters to IOP change were: first applanation pressure (AP1), first applanation velocity (A1 velocity), and first applanation time (A1 time) with r = 0.736, 0.731 and 0.725, respectively, and all of them at significance level of 0.0001 (*p* < 0.0001) referenced by the double asterisk next to the value of r. One of these three corneal response parameters was chosen, along with central corneal thickness (CCT), corneal curvature (R) and corneal material stiffness parameter (µ), to enter an estimation algorithm for IOP. On the other hand, the first applanation length (A1 length) and stiffness parameter (SP-HC) were the most associated response parameters to corneal material change with correlation coefficients of 0.471 and 0.442, respectively, at significance level of 0.01 (*p* < 0.01). 

### 3.3. Clinical Validation of Numerical Results

The clinical dataset for 476 participants from Milan and Rio in Table 1 was used in the validation process. All participants performed the air puff test using the same device (CorVis-ST). The spatial and temporal corneal deformations for three participants are shown in Figure 9 in comparison with the deformations from corneal patient-specific FSI models. A good agreement and close behaviour to the clinical corneal behaviour was achieved. The left column of graphs shows the spatial corneal deformation profiles at four time steps T = 5, 8, 10, 16 ms. The difference in profiles is due to the fact that biological tissue is different in responding to the air puff and it is not guaranteed that the puff is applied to the cornea centre with the same angle and distance from the nozzle. The right column of figures shows the temporal apical deformation numerically and clinically with the value of root mean square error shown on the top.

To validate the parametric study, the same descriptive statistics and correlation analysis, which were done for the parametric study, were performed to the clinical dataset to see if there are any differences, before considering them in the IOP estimation algorithm in the future. Figure 10 provides a bar-chart to compare the means and standard deviations of the dynamic corneal response parameters numerically and clinically. The biggest difference was in the first applanation deformation amplitude with 76.9% higher and HC deformation amplitude with 22.2% lower. In terms of Pearson’s correlations, the clinical dataset showed that the first applanation pressure remained the highest correlated parameter to IOP (r = 0.927, *p* < 0.0001) followed by A1 time (r = 0.889, *p* < 0.0001) and stiffness parameter (SP-HC) (r = 0.857, *p* < 0.0001), which is the same as the numerical database apart from A1 velocity, which was found to be the highest after AP1.

## 4. Discussion

This paper addresses a long-standing challenge of getting accurate IOP measurements in vivo through correcting for the fluid–structure interaction between the air puff and the cornea during non-contact tonometry. There were three main obstacles. The first is the lack of knowing the true loading of the air puff and the true pressure inside the eye. The second is the non-linear behaviour of the corneal tissue in response to these loadings, which until now is hard to predict in vivo to be used in numerical simulations, especially with the dilemma of IOP measurements in vivo being affected by corneal material properties and corneal material estimation being affected by the true loadings on the cornea. The third is the non-linear nature of air jet turbulence at high Reynolds numbers when impinging on the cornea which is not accurate for the air pressure to be put as one formula for all finite element models with different corneal geometrical and material parameters. 

The current study made use of Arbitrary Lagrangian Eulerian (ALE) deforming mesh in numerical simulation of the non-contact tonometry test to couple between the air puff and the eye models. Then, it was used to produce a parametric study making the air velocity the same at the nozzle tip and changing the corneal geometrical parameters (CCT and R), corneal material stiffness (µ) and intraocular pressure (IOP) to better understand the corneal material behaviour under dynamic loading with the final aim to produce IOP and corneal material estimation algorithms considering the fluid–structure interaction from outside the eye. The accurate material characterisation for cornea can assist ophthalmologists and surgeons in treatment, management and surgical planning before any physical intervention with the eye, as well as in diagnosis of some diseases which alter the corneal stiffness such as keratoconus and ectatic diseases [67,68,69]. On the other hand, biomechanical correction of IOP measurement has been the focus of many studies in the past [14,50,70]. Some studies focused on the association of IOP with central corneal thickness CCT and corneal curvature radius R, while others studied the material properties effect, but most of them were structural in nature with no sufficient consideration to the FSI effect between cornea and the air puff. 

The FSI coupled model between the eye and an air puff was successfully built and validated through comparison of the corneal deformations from the numerical model against clinical corneal response parameters acquired from CorVis-ST for in vivo human eyes and compared against finite element models only where ALE deforming mesh was not applied and showed significant improvement in corneal deformations compared to clinical data. This new modelling method will be crucial when simulating patient-specific eye models with abnormal geometries as the pressure distribution will not be uniform and varies with time, which is considered in the literature. The clinical comparisons were presented in two forms. The first was by presenting cornea deformation profiles at four time captures of the test (5, 8, 10, 16 ms) along with the temporal apical corneal deformation, which has shown close but not perfect agreement. This is because soft tissue materials are not easily predicted and represented numerically, which is the reason we suggest a new corneal material estimation algorithm based on fluid–structure interaction. The second was through calculating dynamic corneal response (DCR) parameters (A1 Time (ms), A1 Length (mm), A1 Velocity (mm/s), HC Time (ms), Peak Distance (mm), A1 Deformation Amplitude (mm), HC Deformation Amplitude (mm), AP1 (mmHg) and SP-HC Stiffness parameter) and comparing them against the same parameters obtained clinically. The correlation analyses produced between these parameters were to develop a new corneal material estimation algorithm that depends not only on patient’s age but also on the patient specific corneal response parameters. The air puff was analysed for three main variables: the axial velocity (V3), the parallel velocity (V1) and the pressure (P). These three variables gave indication on the validity of the solution in both models, the CFD and the FE. Values of V3 and V1 validated the near wall treatment of the CFD code, the transport equations’ solution at the impingement region and wall jet region, which is more obvious at (Y/D = 1, 2).

From the results, some limitations appeared when comparing between the numerical and the clinical deformations, as this difference refers to more than one parameter. The first parameter is the boundary conditions applied to the eye model. The eye model was supported from the equatorial nodes to prevent movement in the anterior–posterior direction. There is some work in progress by colleagues to simulate the fatty tissue around the eye to remove that boundary condition and allow whole eye movement. Another important material effect is the hysteresis influence, which is related to the viscoelastic behaviour of the cornea. When the cornea reflects back after the application of the air puff, it has some relaxation time and memory effect to return back to the original geometry. A third parameter is the air puff shooting direction, which can sometimes be at an angle from the eye axis and a modification for the mesh was done to apply the air puff at the same angle as the clinical shooting. A fourth parameter is the sclera dimensions, since we do not have information about the accurate dimensions specific to each patient. Therefore, we used idealised geometry for the sclera based on average clinical dimensions; however, depending on age of the patient, the material stiffness was adjusted. In addition, although we validated our model using healthy clinical geometrical data, our code can work with any geometry even if it is distorted or has any pathological condition including keratoconic eyes. 

## 5. Conclusions

In conclusion, we relied on ALE deforming mesh models to improve consideration of the eye model to the changes in pressure values and distribution of the air puff during the non-contact tonometry test. Following the methodology and after validating the numerical model, we performed a parametric study: To validate the numerical method on a wider range of eye model parameters.To see how air puff pressure value and distribution are affected by corneal parameters change.To know which corneal response parameters are affected by IOP and material changes.To produce estimation algorithms for IOP and corneal material.

This study successfully demonstrated that the use of CFD and ALE deforming mesh are needed to improve the accuracy of interaction between the cornea and air puff tonometer. The use of these methods will also be crucial if patient specific eye models with distorted geometry need to be simulated.

## Figures and Tables

**Figure 1 ijerph-17-00054-f001:**
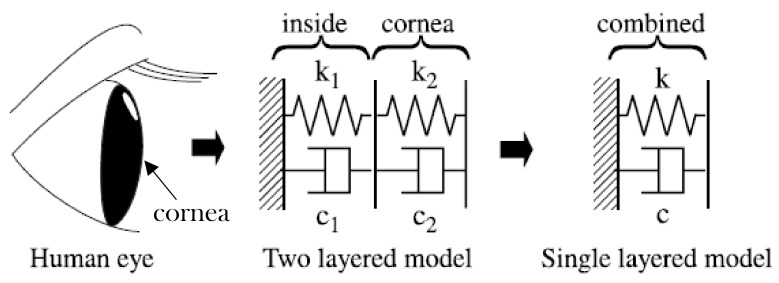
Dynamic model of the human eye as one and two degrees of freedom, considering cornea as one layer and the internal fluids as another layer [14].

**Figure 2 ijerph-17-00054-f002:**
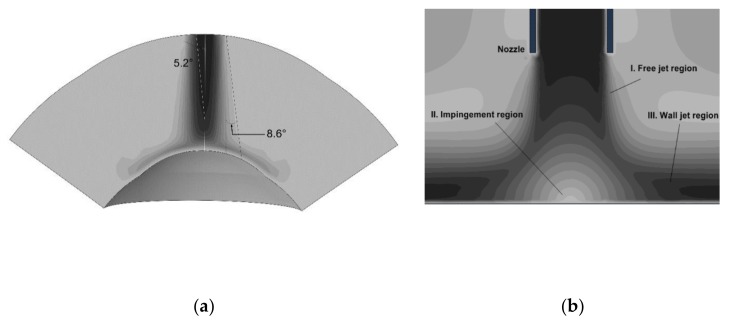
Round impinging jet diffusion (**a**); and the impinging jet different regions (**b**).

**Figure 3 ijerph-17-00054-f003:**
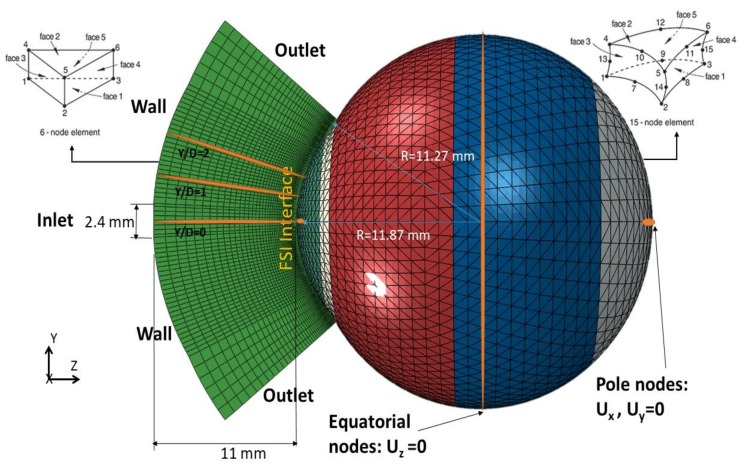
Geometry definition of the air puff and eye domains showing key dimensions, element types and boundary conditions. Ux, Uy and Uz are the deformations in the three dimensions.

**Figure 4 ijerph-17-00054-f004:**
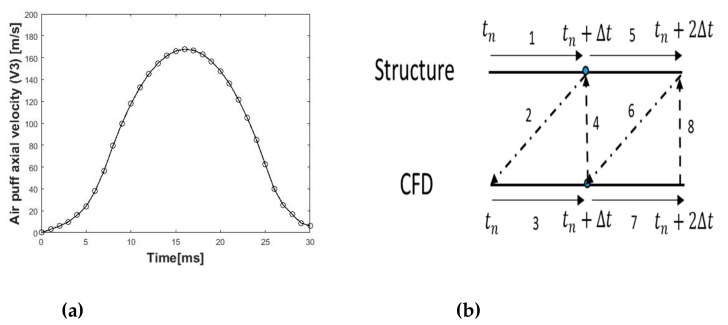
(**a**) Temporal velocity profile at the air puff nozzle fed as inlet boundary condition to the air puff CFD model; and (**b**) flow of the solution in the fluid–structure interaction coupling at each time step.

**Figure 5 ijerph-17-00054-f005:**
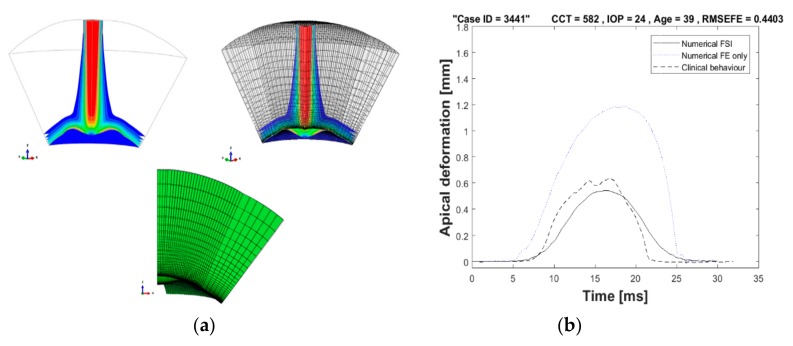
(**a**) Arbitrary Lagrangian–Eulerian (ALE) deforming mesh shown on a quarter model of the air puff; and (**b**) comparison of apical deformation with finite element only models in absence of ALE deforming mesh.

**Figure 6 ijerph-17-00054-f006:**
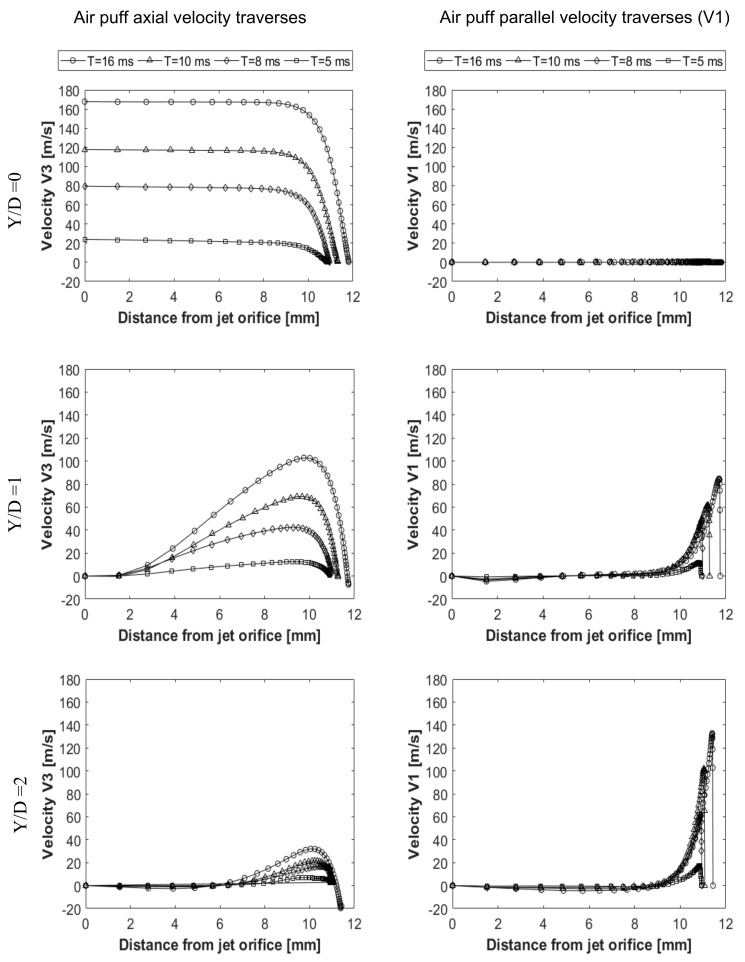
Air puff velocity components (V3 and V1) at axial traverses Y/D = 0, Y/D = 1, Y/D = 2 and 4 time steps at T = 5, 8, 10, 16 ms.

**Figure 7 ijerph-17-00054-f007:**
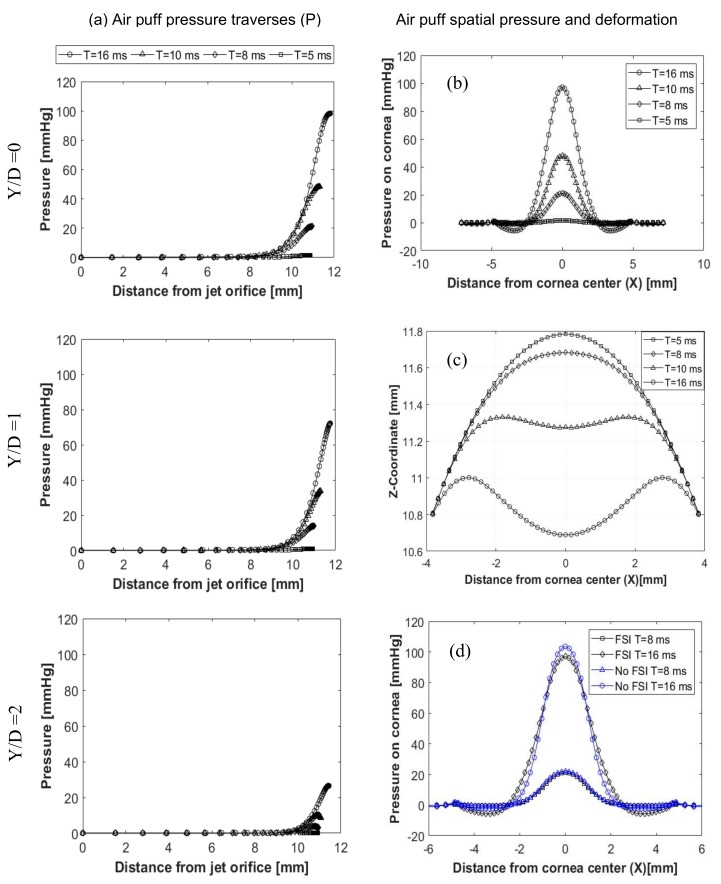
Air puff total pressure traverses (P), at Y/D=0, Y/D=1, Y/D=2 (**a**); spatial pressure distribution on the cornea (**b**); cornea deformation profiles (**c**); and explanation of the FSI effect on the pressure distribution (**d**).

**Figure 8 ijerph-17-00054-f008:**
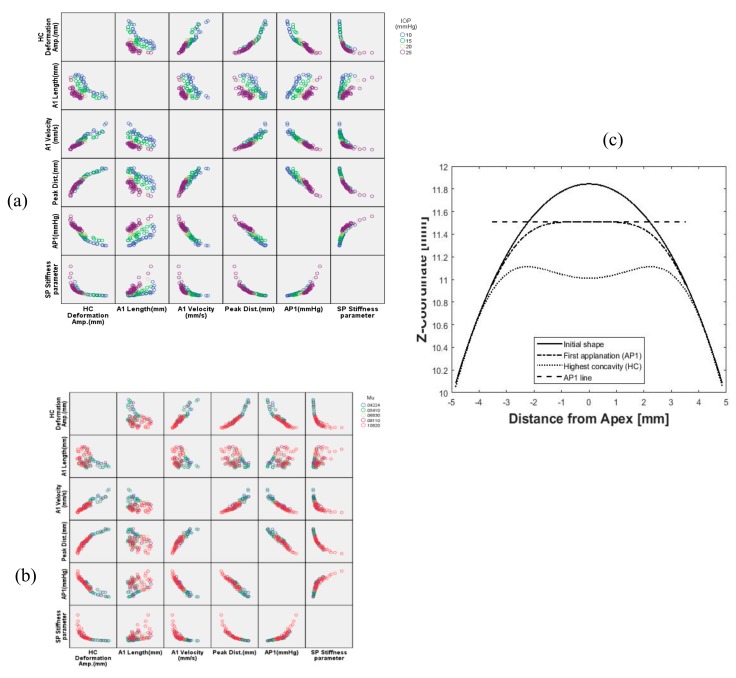
Influence matrix of changing intraocular pressure (IOP) (**a**) and corneal material stiffness (**b**) on corneal response parameters; and (**c**) the corneal profile stages.

**Figure 9 ijerph-17-00054-f009:**
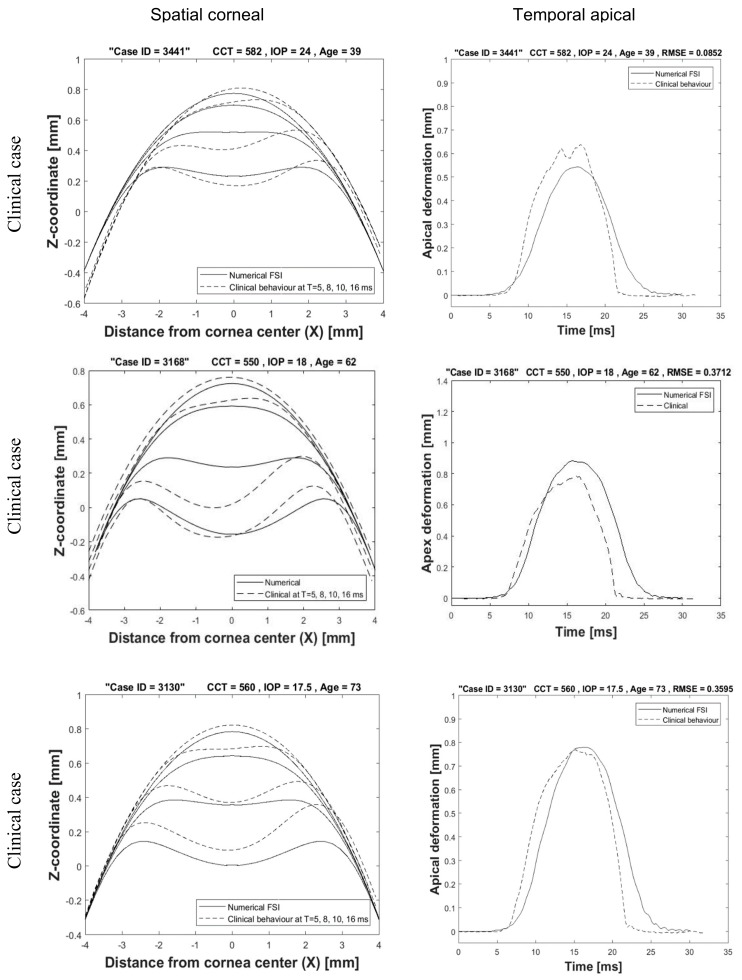
Spatial corneal deformation and temporal apical deformation comparison with three clinical cases.

**Figure 10 ijerph-17-00054-f010:**
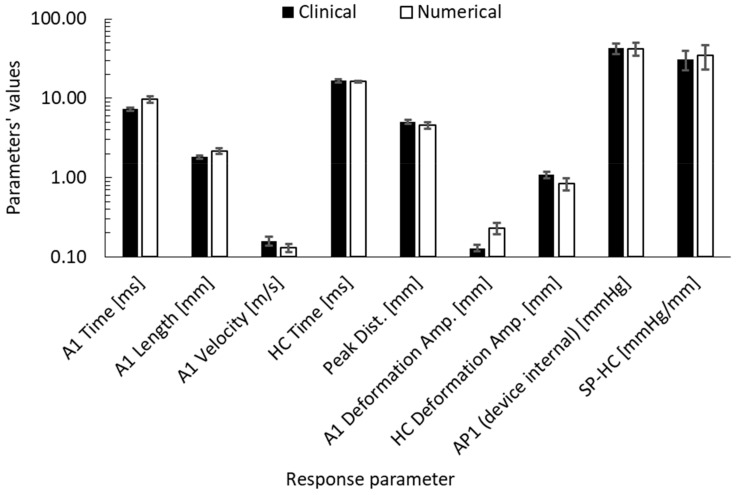
Corneal response parameters’ comparison between clinical and numerical results of the parametric study.

**Table 1 ijerph-17-00054-t001:** Clinical dataset used in validation of the numerical model of the air puff test.

Datasets	Participants	Age (years)	CCT (μm)	CVS-IOP (mmHg)
Dataset 1 (Milan)	225	38 ± 17.2 (7–91)	543 ± 31.5 (458–635)	15.7 ± 2.35 (11–25)
Dataset 2 (Rio)	251	43 ± 16.5 (8–87)	539 ± 33.2 (454–629)	14.8 ± 3.06 (6–34)

Note: CCT, central corneal thickness; CVS-IOP, CorVis IOP measurement.

**Table 2 ijerph-17-00054-t002:** Descriptive statistics of 110 models of the parametric study; the bold line separates input from output parameters.

Variable	Mean	Std. Deviation	Minimum	Maximum
IOP (mmHg)	18.36	6.25	10	25
CCT (µm)	550.45	73.99	445	645
µ	0.0712	0.0236	0.0422	0.1082
R (mm)	7.82	0.33	7.4	8.4
A1 Time (ms)	9.66	0.97	7.81	12.47
A1 Length (mm)	2.15	0.19	1.91	2.62
A1 Velocity (mm/s)	0.13	0.04	0.06	0.21
HC Time (ms)	16.21	0.36	15.3	16.9
Peak Distance (mm)	4.58	0.95	2.46	6.62
A1 Def. Amp. (mm)	0.23	0.05	0.17	0.39
HC Def. Amp. (mm)	0.84	0.3	0.42	1.77
AP1(mmHg)	42.09	12.09	18.82	75.24
SP-HC	34.69	21.92	5	109.59

**Table 3 ijerph-17-00054-t003:** Correlation and relationship significance analysis between input and output parameters of the parametric study.

Variable		A1 Time (ms)	A1 Length mm)	A1 Velocity (mm/s)	HC Time (ms)	Peak Dist. (mm)	A1 Deformation Amp. (mm)	HC Deformation Amp. (mm)	AP1 (mmHg)	SP-HC Stiffness parameter
IOP (mmHg)	Pearson Correlation (r)	0.725 ^**^	−0.455 ^**^	−0.731 ^**^	−0.255 ^**^	−0.616 ^**^	−0.403 ^**^	−0.635 ^**^	0.736 ^**^	0.442 ^**^
	Sig. (two-tailed)	0.000	0.000	0.000	0.007	0.000	0.000	0.000	0.000	0.000
CCT (µm)	Pearson Correlation (r)	0.382 ^**^	0.637 ^**^	−0.206 ^*^	−0.122	−0.500 ^**^	0.673 ^**^	−0.493 ^**^	0.385 ^**^	0.468 ^**^
	Sig. (two-tailed)	0.000	0.000	0.031	0.204	0.000	0.000	0.000	0.000	0.000
μ	Pearson Correlation (r)	0.338 ^**^	0.471 ^**^	−0.367 ^**^	−0.280 ^**^	−0.407 ^**^	0.432 ^**^	−0.377 ^**^	0.355 ^**^	0.434 ^**^
	Sig. (two-tailed)	0.000	0.000	0.000	0.003	0.000	0.000	0.000	0.000	0.000
R (mm)	Pearson Correlation (r)	−0.007	−0.056	−0.067	0.032	0.088	−0.253 ^**^	−0.052	0.007	−0.088
	Sig. (two-tailed)	0.946	0.564	0.486	0.741	0.362	0.008	0.592	0.945	0.362

Note: IOP, intraocular pressure; CCT, central corneal thickness; μ, corneal material stiffness coefficient; R, corneal curvature radius; A1 is the first applanation; HC, highest concavity; AP1, first applanation pressure; SP-HC, stiffness parameter at highest concavity. **: Correlation is significant at the 0.01 level. *: Correlation is significant at the 0.05 level.

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
