# Peer review of "Simulation of Air Puff Tonometry Test Using Arbitrary Lagrangian–Eulerian (ALE) Deforming Mesh for Corneal Material Characterisation"

_ijerph, 2019, doi:10.3390/ijerph17010054_

Round 1
Reviewer 1 Report
This manuscript proposes a new method to improve numerical simulation of the non-contact tonometer test by using Arbitrary Eulerian-Lagrangian deforming mesh in the coupling between computational fluid dynamics model of an air jet and finite element model of the human eye. The methods used in this work include computational fluid dynamics model simulated impingement of the air puff and employed Spallart-Allmaras model to capture turbulence of the air jet. The method validated on 476 healthy patients from two different datasets. In result, the authors claimed that their method can improve the traditional understanding of how pressure distribution on cornea changes with time of the test. The proposed method is technically sound, and the method validation is also reasonable sound. However, several concerns and limitations of this manuscript are listed below:
The method effectiveness is slightly in skepticism. First, the motivation of using ALE deforming meshi method is not clear. If it is about remapping the solution variables to the new meshes is of second-order accuracy and conserves mass and momentum, or adaptive deformable mesh, it will be better to see the ablation study in the experiments. In 3.2 parametric study results, it is unclear how to obtain 110 models in detailed. In addition, the authors did not explain what the major findings are through the results. The writing likes a kind of project report without essential scientific analysis. For clinical dataset description, it is unclear how the patient’s data were collected if there were all healthy patients. The section “3.3 clinical validation of numerical results “should include more details about the patient data, such as healthy status, rather than putting all this critical information in the Ethical statement. The writing is not format well. For 4. Discussion, if the ALE is important part of the method, why did the authors mention it at all in this section? It is better to have more discussion about the major contribution included in this section.
Author Response
Manuscript ID: ijerph-612404
Title: Simulation of Air Puff Tonometry Test Using Arbitrary Lagrangian-Eulerian (ALE) Deforming Mesh for Corneal Material Characterisation
Authors: Osama Maklad *, Ashkan Eliasy, Kai-Jung Chen, Vassilios Theofilis, Ahmed Elsheikh
December 9th 2019,
To the Editor:
Dear respected Reviewer;
We appreciate your comments and suggested revision of our manuscript entitled:” Simulation of Air Puff Tonometry Test Using Arbitrary Lagrangian-Eulerian (ALE) Deforming Mesh for Corneal Material Characterisation”.
Please find below our response to your valued comments and with track changes activated in the revised version of the manuscript. We hope that these answers clear any doubts that you might have.
Yours Sincerely,
Osama Maklad
*Please note that all line numbers mentioned in the responses are when the document on “Simple Markup”
Response to Reviewer 1 comments:
Point 1: Method effectiveness and motivation of using ALE deforming mesh
The method effectiveness is slightly in skepticism. First, the motivation of using ALE deforming meshi method is not clear. If it is about remapping the solution variables to the new meshes is of second-order accuracy and conserves mass and momentum, or adaptive deformable mesh, it will be better to see the ablation study in the experiments.
Response 1:
Dear Reviewer, thank you for reviewing our manuscript and your valuable comments. The main motivation for using ALE deforming mesh is the adaptive mesh which communicates the changes in both simulations (air puff domain and the eye model). Also, it is very important for stability of the fluid model when the boundaries (Cornea & Sclera) are moving and are time dependant. If this method was not applied, the fluid mesh would be distorted and the solution will diverge which was the main reason that we decided to use ALE method. Second-order accuracy and conservation of mass and momentum are necessities for the method to succeed and complete the CFD analysis in the FE platform that we used. We have added a comparison with finite element only model of the eye where ALE was not applied in Figure 5 and lines (244-248, Section 2.3.1).
Point 2: Parametric study: How 110 models were obtained? And what are the major findings of the study?
In 3.2 parametric study results, it is unclear how to obtain 110 models in detailed. In addition, the authors did not explain what the major findings are through the results.
Response 2:
Thank you for the comment and apologies if the explanation was in different parts or missing. The 110 models were obtained by changing four input parameters (IOP, CCT, µ and R) using a bespoke software package to generate the human eye model [35] and newly generated MATLAB® code to build the air domain in front of the cornea (This was added to line 311-313, Section 3.2). Descriptive statistics of the ranges are given in table 2.
In conclusion, we relied on ALE deforming mesh models to improve consideration of the eye model to the changes in pressure values and distribution of the air puff during the non-contact tonometry test. Following the methodology and after validating the numerical model, we performed a parametric study to:
Validate the numerical method on a wider range of eye model parameters. See how air puff pressure value and distribution are affected by corneal parameters change. To know which corneal response parameters are affected by IOP and material changes. To produce estimation algorithms for IOP and corneal material.This study successfully demonstrated that the use of CFD and ALE deforming mesh are needed to improve the accuracy of interaction between the cornea and air puff tonometer. The use of these methods will also be crucial if patient-specific eye models with distorted geometry are required to be simulated. This conclusion part was added to the end of the discussion.
Point 3: Participants’ data collection and clinical validation.
For clinical dataset description, it is unclear how the patient’s data were collected if there were all healthy patients. The section “3.3 clinical validation of numerical results “should include more details about the patient data, such as healthy status, rather than putting all this critical information in the Ethical statement.
Response 3:
Thank you for the comment, “patients” word here wasn’t accurate, we modified it to “participants”. Details of the clinical datasets from both centres are given in Table 1, also it was mentioned in section 2.4 that the inclusion criteria of healthy subjects were a Belin/Ambrósio Enhanced Ectasia total deviation index (BAD-D) of less than 1.6 the standard deviation (SD) from normative values in both eyes, no previous ocular surgery and disease, myopia less than 10D and no concurrent or previous glaucoma or hypotonic therapies. Moreover, to confirm the diagnosis, all exams of each clinic were blindly re-evaluated by a corneal expert at the other clinic.
Point 4: Discussion of the major contribution including ALE effectiveness.
For 4. Discussion, if the ALE is important part of the method, why did the authors mention it at all in this section? It is better to have more discussion about the major contribution included in this section.
Response 4:
Thank you for the comment, the discussion was improved (Lines 400-407 and copied below).
“The coupled model of fluid structure interaction (FSI) between the eye and an air puff was successfully built and validated through comparison of the corneal deformations from the numerical model against clinical corneal response parameters acquired from CorVis-ST for in-vivo human eyes and compared against finite element models only where ALE deforming mesh was not applied and showed significant improvement in corneal deformations compared to clinical data. This new modeling method will be crucial when simulating patient-specific eye models with abnormal geometries as the pressure distribution will not be uniform and varies with time which was considered in the literature.”

Reviewer 2 Report
The strength of the manuscript is in the use of CFD and the elaborate useof eye details in the modeling of tonometry process.
The analysis of the air puff for observing the effects of velocity,
pressure and mesh deformation was done nicely and the results look sound and
reasonable.
A comprehensive simulation has been performed on non-contact tonometry using CFD. The manuscript summarizes the finding well.
Author Response
Response to Reviewer 2 comments:
Point 1: A comprehensive simulation has been performed on non-contact tonometry using CFD. The manuscript summarizes the finding well.
Response 1:
Thank you for the comment.
Reviewer 3 Report
The paper is generally well written and is accessible for a wide range of researchers. However, there a several punctuation mistakes (missing space, or too many spaces, commas missing).
The in-text citation does not follow the journals requirements (please see https://www.mdpi.com/journal/ijerph/instructions#preparation) and when naming papers within a sentence, only the surname should be used.
Some abbreviations are used differently across the text (e.g. 1DOF/ 1-DOF).
Line 44: In most of the tonometer measurements nowadays it is not known which force / pressure is applied to the cornea. Only Goldmann tonometry relates the force to the IOP. in both devices assessed in this paper, the pressure/force is not known and is not related to the the IOP.
Figure 1: It would be good to label the eye, to make it clear for non-clinicians which parts of the eye are related to within the models.
Methods: It is not really clear how the scleral data was determined by using the Pentacam. Also, it would have been good, to have a clear rational for the ring segment choice. It remains completely unclear how scleral data was obtained. The author states that this is a patient-specific modelling approach, but from the methods only (limited!) corneal data was available.
Ref 44: One of the fundamental assumptions (fluid dynamic) is based on a 3d model derived from an enucleated eye with the aim to produce a glaucoma model. What is the authors rational behind using those parameters?
Figure 4: Those pressure curves are from within the piston chamber not on the eye plane. It is not clear how this effects the outcome of the model as the applied pressure on the cornea is far less.
There is a contradiction about the patients-data (line 186-187, 243).
What were the inclusion and exclusion criteria of those patients? If they are considered as patients, how can they be deemed as healthy (firstly it can be assumed that they are patients as they were in hospital and secondly it is not clear how systematic conditions (diabetes etc.) were controlled or excluded)? It is known that biomechanics properties are effected by a number of conditions which are not assessable by just using a video topographer. The demographic but also the clinical parameters are widespread, it would be good to include a table with patients data.
Results line 341: Was the same device transported from clinic to clinic? Which software version was used?
The analyses from line 355 to 358 does not make sense as the parameters are derived from another.
The author should discuss limitation and assumption of the model in depth at the end of the paper. This is completely missing.
Author Response
Manuscript ID: ijerph-612404
Title: Simulation of Air Puff Tonometry Test Using Arbitrary Lagrangian-Eulerian (ALE) Deforming Mesh for Corneal Material Characterisation
Authors: Osama Maklad *, Ashkan Eliasy, Kai-Jung Chen, Vassilios Theofilis, Ahmed Elsheikh
December 9th 2019,
To the Editor:
Dear respected Reviewers;
We appreciate your comments and suggested revision of our manuscript entitled:” Simulation of Air Puff Tonometry Test Using Arbitrary Lagrangian-Eulerian (ALE) Deforming Mesh for Corneal Material Characterisation”.
Please find below our response to your valued comments and with track changes activated in the revised version of the manuscript. We hope that these answers clear any doubts that you might have.
Yours Sincerely,
Osama Maklad
*Please note that all line numbers mentioned in the responses are when the document on “Simple Markup”
Response to Reviewer 3 comments:
Point 1: Punctuation mistakes
There a several punctuation mistakes (missing space, or too many spaces, commas missing).
Response 1:
Dear Reviewer, thank you for reviewing our manuscript and your valuable comments. We revised the whole manuscript and corrected these mistakes.
Point 2: In-text citations
The in-text citation does not follow the journals requirements and when naming papers within a sentence, only the surname should be used.
Response 2:
Thank you for the comment. We have revised the in-text citations and referencing scheme in the whole paper.
Point 3: Abbreviations consistency
Some abbreviations are used differently across the text (e.g. 1DOF/ 1-DOF).
Response 3:
We have revised the abbreviations in the whole paper and made sure they were consistent.
Point 4: Which force / pressure is applied to the cornea?
Line 44: In most of the tonometer measurements nowadays it is not known which force / pressure is applied to the cornea. Only Goldmann tonometry relates the force to the IOP in both devices assessed in this paper, the pressure/force is not known and is not related to the IOP.
Response 4:
We agree with your comment and have modified the sentence accordingly (Line 43, Section 1). It was one of the main motivations of this paper to conduct high fidelity simulation of the air puff using CFD to know the distribution of the dynamic pressure (which is a function of the air velocity) acting on the cornea. In previous studies [6,19], it was experimentally estimated that the pressure at the nozzle of the device reduced by 50% at the time of reaching the cornea. However, this was only an estimation and the distribution of pressure over time which is dependent on the corneal shape had not been simulated before. Once the pressure on the cornea was known in value and distribution, the corneal deformation parameters were related to the IOP and used to produce an estimation algorithm.
Point 5: Figure 1
Figure 1: It would be good to label the eye, to make it clear for non-clinicians which parts of the eye are related to within the models.
Response 5:
Thank you for your comment, we have added a label to the eye in figure 1 and modified the figure caption.
Point 6: Methods: Scleral data using Pentacam and patient-specific modelling
It is not really clear how the scleral data was determined by using the Pentacam. It remains completely unclear how scleral data was obtained. The author states that this is a patient-specific modelling approach, but from the methods only (limited!) corneal data was available.
Response 6:
Thank you for the comment, sorry for the confusion. In this study, which is building on our previous numerical models [36], we use corneal patient-specific only in our models. For sclera we do not have information about the accurate dimensions specific to each patient. Hence, we use idealised geometry for the sclera based on average clinical dimensions, however, depending on age of the patient, the material stiffness was adjusted. In the methodology we added a sentence (Lines 137-139, Section 2.1) that by patient specific eye models we mean the material of the whole eye globe is specific to the patient and adjusted based on age. However, the geometry of sclera is not specific to the patient and is relying on average values [36].
Point 7: Methods: ring segment choice
Also, it would have been good, to have a clear rational for the ring segment choice.
Response 7:
This choice of number of rings was the result of a mesh independence study and the corneal deformations and pressure on the cornea stabilised at this number of rings. We have added a sentence (Lines 132-133, Section 2.1) to provide Figure 1 below for the mesh independence study as a supplementary material.
Point 8: Density and bulk modulus for the fluid cavity
Ref 44: One of the fundamental assumptions (fluid dynamic) is based on a 3d model derived from an enucleated eye with the aim to produce a glaucoma model. What is the authors rational behind using those parameters?
Response 8:
These properties are the density and compressibility modulus (Bulk modulus) for water and they were used as approximate values for aqueous humour in line with previous studies [45]. These parameters are input values for the fluid cavity inside the eye.
Point 9: Figure 4
Figure 4: Those pressure curves are from within the piston chamber not on the eye plane. It is not clear how this affects the outcome of the model as the applied pressure on the cornea is far less.
Response 9: Figure 4 shows the air velocity on the cornea not the pressure, as the pressure distribution is unknown and will be the outcome of the CFD model. We have conducted another CFD model for the CorVis-ST from inside the piston chamber all the way to the exit nozzle to make sure the information about the air velocity at the nozzle, provided by the manufacturer, is correct. Please see Figure 2 below. We have added this figure to the supplementary material.
Point 10: Contradiction about the patients’ data
A There is a contradiction about the patients-data (line 186-187, 243).
Response 10:
The text in lines (186-187, Section 2.2) covers the numerical model and how it’s flexible to read the Pentacam files for topographical cornea patient-specific eye models and apply the CFD air puff on them. While in line (243, Section 2.4) we were talking about the clinical data of healthy participants used in validation of the numerical results. Although, we validated our model on healthy clinical data, our code can work with any geometry even if it is distorted or has any pathological condition including keratoconic eyes, but this is considered in our future work. I have added this point to the discussion to make that clear (Lines 433-437, Section 4).
Point 11: Clinical data
What were the inclusion and exclusion criteria of those patients? If they are considered as patients, how can they be deemed as healthy (firstly it can be assumed that they are patients as they were in hospital and secondly it is not clear how systematic conditions -diabetes etc.- were controlled or excluded)? It is known that biomechanics properties are affected by a number of conditions which are not assessable by just using a video topographer. The demographic but also the clinical parameters are widespread, it would be good to include a table with patients’ data.
Response 11:
It was mentioned in section 2.4 that the inclusion criteria of healthy subjects were a Belin/Ambrósio Enhanced Ectasia total deviation index (BAD-D) of less than 1.6 the standard deviation (SD) from normative values in both eyes, no previous ocular surgery and systematic conditions including diabetes, myopia less than 10D and no concurrent or previous glaucoma or hypotonic therapies. Moreover, to confirm the diagnosis, all exams of each clinic were blindly re-evaluated by a corneal expert at the other clinic. Thank you for pointing out the “patients” matter, this was modified to “participants” in the whole paper.
Point 12: Device used for clinical data
Results line 341: Was the same device transported from clinic to clinic? Which software version was used?
Response 12:
The clinical data were collected at two clinical centres, using two different devices and by different technicians. However, all exams were blindly re-evaluated by a corneal expert at the other clinic. The software version used in both centres was (1.2r1307), this was added to the manuscript (Line 255, Section 2.4).
Point 13: Analyses from line 355 to 358
The analyses from line 355 to 358 does not make sense as the parameters are derived from another.
Response 13:
A1 deformation amplitude is different than HC deformation amplitude (A1 stands for first applanation and HC stands for highest concavity) and they are not derived from each other [2,13,35]. That’s why we are suggesting a change to the numerical material parameters to correct for the corneal response to the air puff which was not considered before this work.
Point 14: Discussion of limitations and assumptions
The author should discuss limitation and assumption of the model in depth at the end of the paper.
Response 14:
As per your comment, we have added a paragraph in the discussion detailing the study limitations (Lines 422-437, Section 4).

Round 2
Reviewer 1 Report
Thanks for your response to my comments. Based on revision, it is good to go. This work has some potential to go further.
Reviewer 3 Report
The new version looks much better. However, there are two sections
called '4. Discussion' and there are still some minor punctuation
and grammar mistakes as well as missing spaces between words.